# The Role of the Gut Microbiome in the Development of Acute Pancreatitis

**DOI:** 10.3390/ijms25021159

**Published:** 2024-01-18

**Authors:** Ruilin Zhou, Qingyang Wu, Zihan Yang, Yanna Cai, Duan Wang, Dong Wu

**Affiliations:** 1Department of Gastroenterology, Peking Union Medical College Hospital, Chinese Academy of Medical Sciences and Peking Union Medical College, Beijing 100730, China; zhouruilin@pumch.cn (R.Z.); pumchyzh@student.pumc.edu.cn (Z.Y.); s2023001077@pumc.edu.cn (Y.C.); 2Eight-Year Medical Doctor Program, Chinese Academy of Medical Sciences and Peking Union Medical College, Beijing 100730, China; qy-wu17@student.pumc.edu.cn (Q.W.); wangd17@student.pumc.edu.cn (D.W.); 3Clinical Epidemiology Unit, Peking Union Medical College Hospital, Chinese Academy of Medical Sciences and Peking Union Medical College, Beijing 100730, China

**Keywords:** gut microbiota, acute pancreatitis, dysbiosis of intestinal flora, intestinal barrier dysfunction

## Abstract

With the explosion research on the gut microbiome in the recent years, much insight has been accumulated in comprehending the crosstalk between the gut microbiota community and host health. Acute pancreatitis (AP) is one of the gastrointestinal diseases associated with significant morbidity and subsequent mortality. Studies have elucidated that gut microbiota are engaged in the pathological process of AP. Herein, we summarize the major roles of the gut microbiome in the development of AP. We then portray the association between dysbiosis of the gut microbiota and the severity of AP. Finally, we illustrate the promises and challenges that arise when seeking to incorporate the microbiome in acute pancreatitis treatment.

## 1. Introduction

Acute pancreatitis (AP) is a type of inflammatory response in which pancreatic enzymes are activated in the pancreas due to multiple etiologies, causing self-digestion, edema, hemorrhage and even necrosis in the pancreatic tissue [1,2]. AP occurs due to various causative factors, such as alcohol, infection, obstruction and intestinal microecological factors. The intestines are connected to the outside world and need to fight against the invasion of outside bacteria and other pathogens, and at the same time, the intestines contain many colonizing bacteria. The immune system monitors and responds to the intestinal microbial consortium with both tolerance and intolerance. The pancreas maintains close communication with the digestive tract through the pancreatic ducts. Intestinal dysfunction is a common complication of severe acute pancreatitis (SAP) and an important factor contributing to the development of the disease and even inducing multi-organ dysfunction [3,4]. Dysbiosis of the intestinal flora may play an important role in the pathogenesis of AP and affect the prognosis, including structural disorders of the intestinal flora and bacterial translocation [5]. It may also affect host metabolism, increase the production of toxic metabolites and affect the treatment of AP. Intestinal dysfunction includes intestinal mucosal barrier damage and intestinal dysmotility. In recent years, there has been increasing evidence that the intestinal flora plays an important role in the progression of AP and that the balance of intestinal microecology can reduce bacterial overgrowth and the occurrence of intestinal-derived infections by regulating the immune system and metabolic pathways and forming a biological barrier in the intestinal mucosa. Moreover, the resumption of feeding in AP patients remains controversy. Some doctors believe AP should be managed with aggressive hydration with intravenous fluids and fasting, while other doctors believe oral feeding should be recommenced in mild pancreatitis once pain and nausea and vomiting have resolved. This controversy is focused on the relationship between gut microbes and AP. The doctors insisting on fasting believe the occurrence of AP is closely correlated with the disruption of commensal microbes in the gut and colonization of pathogens. The doctors advocating for early oral feeding believe the recovery of intestine function will aid the reconstruction of gut microbiota and thus promote the recovery of health. Thus, understanding the complexity and molecular aspects of the link between gut microbes and health will help lay the groundwork for new therapies that have been developed [6].

## 2. Dysbiosis of Intestinal Flora in Acute Pancreatitis

The occurrence of AP is inextricably linked to intestinal dysbiosis [6]. There are more than 1500 species of bacteria in the human intestinal tract, and the intestinal flora is characterized by bacteria of varying dynamics, number, variety and complexity. Intestinal microbiota include beneficial bacteria, intermediate or conditionally pathogenic bacteria and harmful bacteria. The interdependence and confrontation among these three types of bacteria maintain the stability of the intestinal microenvironment. During the development and progression of AP, abnormal secretion of trypsin and destruction of the pancreatic structure lead to abnormal pancreatic secretion, which in turn leads to an imbalance of homeostasis in the body and changes in the intestinal flora. It has been shown that intestinal flora can regulate intestinal mucosal barrier function by affecting intestinal mucosal epithelial cell renewal, intestinal permeability, release of intestinal antimicrobial peptides and the intestinal mucus layer. Zhu et al. found that the rate of bacterial detection was proportional to the severity of the patients’ disease [7]. It was shown that *Escherichia coli*, *Enterococcus*, *Enterobacter*, *Immunobacterium*, *Shigella fowleri* and *Bacillus coagulans* were the main intestinal pathogenic bacteria detected in peripheral blood. *Escherichia coli*, *Enterococcus* and *Enterobacteriaceae* are documented as the main pathogens of secondary intestinal infections caused by AP [8]. It was reported that in AP, an imbalance exists in the flora ratio [9,10]. Intestinal aerobic bacteria showed a significant increase in their proportion, represented by *Escherichia coli*, *Enterococcus*, *Enterobacter* and *Streptococcus*, while anaerobic bacteria showed inhibition in growth, represented by *Bifidobacterium*, *Bacteroidetes* and *Prevotella* [6]. The increase in pathogenic bacteria could disrupt the intestinal barrier and increase the intestinal permeability and bacterial translocation. Increased intestinal permeability leads to the occurrence of pathogen-associated molecular patterns (PAMPs) in the blood circulation, activating the innate immune response. The trigger of inflammation further promotes the development of AP. These studies suggest that the disturbance of the intestinal flora is closely associated with AP.

The disturbance of the intestinal flora is associated with the disruption of the intestinal mucosal barrier. The disruption of the intestinal barrier contributes to the development of AP. A normal intestinal mucosal barrier consists of mechanical, chemical, immune and biological barriers. Studies have shown that the intestinal microbiota can regulate the intestinal mucosal barrier function by affecting intestinal epithelial cell regeneration, intestinal permeability, intestinal antimicrobial peptide release and intestinal mucosal layers [11,12,13,14]. There are many reasons for the disruption of the intestinal mucosal barrier. Fasting, gastrointestinal decompression and other therapeutic measures commonly used in the treatment of AP can cause intestinal flora disruption [15,16]. Additionally, microcirculatory disorders caused by reduced blood volume can lead to intestinal ischemia and hypoxia, thus weakening intestinal dynamics [17]. All the above can affect the normal excretion of harmful substances and thus leads to dysbiosis of the intestinal flora. The simultaneous use of a large amount of fluids during treatment can lead to intestinal reperfusion, resulting in intestinal wall cell dysfunction, increased intestinal ischemia–reperfusion injury, reduced intestinal barrier function and disruption of the intestinal microecological balance [18]. During the treatment of pancreatitis, broad-spectrum antibiotics are often used to control the infection, resulting in the suppression of normal intestinal flora [19]. In addition to these aforementioned factors affecting the intestinal mucosa, the role played by intestinal dysbiosis should not be underestimated. Intestinal commensal microbes themselves form a direct intestinal barrier by competing for space and nutrition, resisting colonization by pathogens [20]. Moreover, commensal microbes also secrete short-chain fatty acids, which modulate gut inflammation. When the intestinal mucosal barrier is disrupted, even health-promoting bacteria can weaken host health and promote pancreatitis. Due to the lack of inhibition of the normal flora, fungi and drug-resistant pathogens in the intestine are more likely to multiply [21]. The imbalance of flora species and ratios disturbs the environmental microecological balance of the intestine [22]. All of the above therapeutic measures lead to further dysbiosis and translocation of intestinal ecology, the increased release of endotoxins and increased inflammatory response in patients with AP. These studies suggest that the disturbance of the intestinal flora is closely correlated with AP.

## 3. Effect of Intestinal Flora on Acute Pancreatitis

An increase in pathogenic bacteria and a decrease in beneficial bacteria have been observed in AP patients after gut microbiota dysbiosis [23]. The beneficial bacteria that have been reported include *Bacteroidetes*, *Lactobacillus* and *Bifidobacterium*. These beneficial bacteria contribute substantially to the health status of the host by inhibiting the growth of harmful bacteria and secreting substances essential for gut health [24,25,26]. The pathogenic bacteria that have been reported include *Enterobacteriaceae*, *Clostridia* and *Bacilli*. These pathogenic floras also participate in relevant pathways in the development of AP. The ratio of beneficial bacteria to pathogenic bacteria maintains the normal functioning of the intestine. This ratio is significantly altered in pancreatitis. It was reported that the number of *Escherichia*/*Shigella* increased in rats with pancreatitis, and further research revealed the bacteria aggravated pancreatitis through targeting intestinal epithelial cells and being translocated by activated regulatory T cells, which causes necrosis of the pancreas [27]. Moreover, the elevation of *Escherichia*/*Shigella* increases the abundance of Toll-like receptor 4 (TLR4)/MyD88/p38 mitogen-activated protein (MAPK) and leads to endoplasmic reticulum stress (ERS) signaling-induced intestinal epithelial injury [28]. The TLR4/MyD88/p38 MAPK pathway is associated with tight-junction proteins. ERS signaling is not only associated with tight-junction proteins but also inflammation. Among these molecules, TLR4 plays a role in the pathogenesis of AP [29]. In a nutshell, the destruction of intestinal epithelial cells further causes necrosis of the pancreas.

Gut microbiota not only directly promote AP but also indirectly trigger relevant inflammation and are associated with the severity of AP. The diaminopimelic acid (DAP)-containing bacteria *Romboutsia* and *Allobaculum* are increased in SAP rat models. The increasing abundance of DAP activates the NOD1/RIP2 inflammatory signaling pathway and affects the systemic inflammatory response [30]. Gut-microbiota-derived nicotinamide mononucleotide alleviates AP by being transported to pancreas and converted into NAD^+^, which further activates pancreatic SIRT3 signaling through the SIRT3-PRDX5 pathway. SIRT3 improves the oxidative damage and inflammation caused by AP and then acetylates PRDX5, which aggravates the cell injury of pancreatic acinar cells [31]. *Bifidobacterium* spp. and their metabolite lactate can protect against multiple-organ dysfunction syndrome in AP patients via inhibition of pancreatic and systemic inflammatory responses [32]. Higher abundances of *Proteobacteria* phylum, *Enterobacteriaceae* family, *Escherichia-Shigella* genus and *Klebsiella pneumoniae* but lower abundances of *Bifidobacterium* genus were found in AP patients with acute respiratory distress syndrome [33].This association between gut microbiota and acute respiratory distress syndrome may be related to the lung microenvironment [33]. There are other reports demonstrating an association between specific intestinal floras and the development of acute pancreatitis, as achieved through specialized molecular mechanisms. According to Yue’s research, the decreasing abundance of Lactobacillus is associated with the downregulation of Paneth cells, which are essential components of the intestinal epithelium. Interestingly, the upregulation of Paneth cells could alleviate AP, thus indicating that the loss of Lactobacillus can indirectly exacerbate pancreatic damage [34]. This research also indicated the essential regulatory role of gut microbiota in intestinal epithelial cells and the development of AP. More research indicated the crosstalk between gut microbiota and intestinal epithelial cell receptor activation in the development of AP. It was identified that there existed a bidirectional modulation between the gut microbiota and NLRP3 in the progression of AP, which suggests the interplay of the host and microbiome during AP [35]. The researchers observed synchronous changes in gut microbiota restoration and intestinal NLRP3 inflammasome inactivation during AP recovery.

## 4. Molecular Actors in Acute Pancreatitis

Besides the intestinal flora itself, its metabolites have an effect on intestinal barrier function in AP. SCFAs play an important role in intestinal barrier function. SCFAs are metabolites produced by gut microbiota. SCFAs are a fermentative product of gut bacteria [6]. There are many bacteria that produce SCFAs, including *Blautia hydrotrophica*, *Akkermansia muciniphila*, *Bacteroides vulgatus*, *Coprococcus catus*, *Megasphaera elsdenii*, *Ruminococcus bromii*, *Clostridium butyricum*, *Roseburia inulinivorans* and so on. According to some research, SCFAs have been shown to maintain the mechanical barrier, enhance the immune barrier and regulate the biological barrier of the intestinal mucosa [36,37,38]. Mechanically speaking, SCFAs play crucial roles in the growth of intestinal epithelial cells and the expression of Zo-1 and Occuludin, which are tight-junction proteins in the intestinal epithelium. Intestinal epithelial cells can produce antimicrobial peptides, which participate in the immunity of the intestine. The pH of the gut can also be reduced by SCFAs, leading to an increase in beneficial bacteria. Recent studies have also shown that SCFAs play vital roles in SAP-associated lung injury. The possible mechanisms underlying this include inhibition in the proliferation of pathogens, anti-inflammatory effects, enhancement of intestinal epithelial barrier, a reduction in bacterial translocation and immunomodulatory effects [6].

Bile acids (BAs) and lipopolysaccharides (LPSs) are two more common metabolites closely associated with AP. BAs are also important microbiota metabolites that affect the intestinal barrier and important cholesterol metabolites that can solubilize dietary lipids. Through their antimicrobial activity and by activating host signaling pathways that maintain gut homeostasis, BAs can shape the microbiota composition [39]. The dysbiosis of gut microbiota caused by bile acids leads to the dysregulation of epithelial transport and barrier function, which interacts with the pathogenesis of pancreatitis [40]. BAs are the leading cause of AP, and research has been carried out exclusively on the retrograde infusion of bile acids into the pancreatitis duct. However, systemic bile acids affect the severity of acute pancreatitis under different disease conditions. Research has found that hydrophobic BAs aggravate AP when AP is independent of serum cholecystokinin (CCK). However, BAs reverse CCK-induced injury based on an interaction with the CCK receptor on acinar cells [41,42]. Another metabolite worth mentioning is lipopolysaccharide (LPS). LPSs are an important component of the outer membrane of gram-negative bacteria. According to Vonlaufen’s research, it is believed that LPSs activate the Toll-like receptor 4 (TLR4) and CD 14 pathways of innate immunity, leading to pancreatic damage [43].

Some molecular actors affect the occurrence and development of AP by affecting the gut microbiota. The endogenous cannabinoid system (ECS) is widely expressed in the human body and plays important roles in gastrointestinal functions. Interaction between the ECS and intestinal flora regulates the permeability of the intestinal epithelial barrier [44]. It is reported that the increasing number of ligands and receptors in the ECS in the pancreas is associated with AP. The underlying mechanism is the ECS’s association with the brain–gut–microbiota axis and effect on systemic inflammation [45]. Cannabinoids (CBs) exert their activities of regulating inflammation and gut–adipose tissue signaling through binding to the CB1 and CB2 receptors and non-CB1/non-CB2 receptors such as G-protein-coupled receptor 55 (GPR55) or the transient receptor potential channels [46]. These receptors inhibit adenylate cyclase and the production of cAMP, attenuating the protein kinase A pathway, and thus regulate the inflammation of pancreas.

These newly discovered molecules provide potential therapy targets (Table 1). TLR4 mediates the recognition of bacterial LPSs and is thought to be highly relevant to systemic inflammatory response syndrome. TLR4 has been widely used as a potential mechanistic target for the treatment of AP [34]. SAP stimulated by caerulein (HY-A0190) and LPSs can be reduced by Sitagliptin [47]. Biliary pancreatitis is also one of the main causes of AP. The proposed mechanism is the reflux of bile acids into the pancreatic duct. The increase in bile acid concentration causes an increase in cytoplasmic calcium and further causes damage to pancreatic acinar cells, so targeting bile acid is also a potential treatment method necessitating further research [48]. In addition, NLRP3 was reported to play an essential role in promoting inflammation in AP. NLRP3 is one of the NLR proteins, mediating caspase-1 activation and the secretion of the pro-inflammatory cytokine IL-1β in response to microbial infection and cellular damage [35]. NLR3 also showed a bridging role in gut microbiota and AP. Recently, it has been reported that gut microbiota can secret nicotinamide mononucleotide, which alleviates AP by activating pancreatic SIRT3 signaling [31]. Further studies revealed that during the development of acute pancreatitis, microbiota dysbiosis is promoted by inflammatory factors. Gut microbiota dysbiosis in turn ameliorates the severity of AP, including mitochondrial dysfunction, oxidative damage and inflammation. It was proven that nicotinamide mononucleotide mitigates AP-mediated mitochondrial dysfunction, oxidative damage and inflammation by increasing pancreatic NAD+ levels. Molecular lab tests indicated that gut-microbiota-derived nicotinamide mononucleotide alleviates the severity of AP by activating the SIRT3-PRDX5 pathway. In conclusion, the gut microbiome influences AP in multiple ways (Figure 1).

## 5. Future and Prospects

With an enhanced understanding of gut microbiota and pancreatitis, the treatment of pancreatitis has been constantly improving. The application of probiotics and the early recovery of diet are considered important in the recovery of AP. Probiotics are beneficial bacteria that can be consumed from various sources and can improve health in different ways. In inflammatory diseases, probiotics have many functions. First of all, probiotics compete with harmful bacterial communities to inhibit the growth of pathogenic bacteria [55]. Secondly, probiotics stimulate the intestinal mucosa to secrete mucin, inhibit the binding of pathogens to the intestinal epithelium and reduce their passage through the intestinal wall into peripheral organs [56]. Thirdly, probiotics promote the biosynthesis of glutathione, improve the dysfunction of the intestinal mucosal barrier, reduce oxidative stress, reduce the permeability and apoptosis level of the ileal mucosa and thereby inhibit the migration of intestinal bacteria [57]. Fourthly, probiotics affect the function of immune cells in various ways (for example, by regulating the production of pro-inflammatory and anti-inflammatory cytokines) [58]. Thus, the application of probiotics in AP can restore the intestinal microbiota balance and prevent bacterial translocation and infection. In pancreatitis, the most commonly used probiotics are *Lactobacillus acidophilus*, *Lactobacillus bulgaricus* and *Bifidobacterium bifidum* [59,60,61]. Studies have shown that probiotics may shorten the length of hospital stay and reduce the infection rate in mild AP [62]. Moreover, it has been reported that probiotic supplementation for experimental AP did not show an adverse effect on mortality but showed a positive effect of reduced histopathology of the pancreas and bacterial translocation to the pancreas and mesenteric lymph nodes [63]. In addition, a study by Hooijmans et al. also revealed that single-strain probiotic supplementation might be more effective in reducing the risk of bacterial translocation than multi-strain supplementation. The safety of probiotics is also monitored. The probiotics used in clinical trials are various strains of bacteria that are intended to modulate the gut microbiota and enhance the host immune system, such as *Lactobacillus paracasei* ssp. *paracasei F19* and *Bifidobacterium lactis Bb12*. They have been tested for different conditions, such as infectious diseases and inflammatory bowel diseases [64]. The safety of probiotics clinical trials is generally good, but there are some potential risks and adverse effects that need to be monitored. These include bacteremia, fungemia, endocarditis, sepsis, organ failure, antibiotic resistance and immune dysregulation [65]. However, these adverse events are only documented in case reports of individuals. We lack specific statistics of the adverse effects of probiotics.

Since gut microbiota are closely associated with the development of AP, gut microbiota may have predictive values in AP [66]. With the emergence of the concept of precision medicine, the study of biomarkers has become particularly important. Precision medicine is an emerging approach to disease prevention and management that takes into account differences in an individual’s genes, environment and lifestyle habits [67,68]. Precision medicine is a new medical concept and medical model based on individualized medicine and developed with the cross-application of bioinformatics and big data science [69,70]. The essence of precision medicine is to analyze, identify, validate and apply biomarkers to a large sample of people and specific disease types through analyzing the genome and proteome and other genomics technologies and cutting-edge medical technologies, so as to accurately search for the causes of diseases and the targets of treatments and accurately categorize the different states and processes of a disease, and ultimately to achieve the goal of personalized and precise treatments for diseases and specific patients and to improve the effectiveness of disease diagnosis and treatment as well as prevention. All in all, this improves the efficiency of disease diagnosis and treatment and prevention. AP is categorized as mild or severe, with or without necrosis. For different types of pancreatitis, the prognosis and treatment are very different. For instance, acute necrotizing pancreatitis has a higher mortality and worse outcome than non-necrotizing pancreatitis. Current research suggests that changes in the intestinal flora have a close relationship with the severity and prognosis of pancreatitis (Figure 2). Therefore, the intestinal flora has the potential to become a marker for measuring the prognosis of pancreatitis, thus realizing the precise treatment of pancreatitis. Studies show that the intestinal microbiota in necrotizing pancreatitis patients are distinct from those in non-necrotizing pancreatitis patients’ intestinal microbiota [66]. Necrotizing pancreatitis patients presented a lower abundance of *Bifidobacterium* and *Blautia* and higher abundance of *Enterococcus* and *Escheichia*/*Shigella* compared with non-necrotizing pancreatitis patients. It is worth mentioning that *Bifidobacterium* and *Blautia* are considered probiotic candidates, while *Enterococcus* and *Escheichia*/*Shigella* are regarded as possible pathogens. Surprisingly, both necrotizing pancreatitis and non-necrotizing pancreatitis are associated with a decrease in certain probiotic microbes and increase in opportunistic pathogens. This research also pointed out that the gut microbiota’s influence on the progression of the narcotizing of the pancreas may be associated with ketone body or benzoate metabolism [71]. Thus, *Enterococcus faecium* and *Finegoldia magna* are potential biomarkers for necrotizing pancreatitis [66]. Research by others showed that different types of pancreatitis are associated with different gut microbiota compositions. The gut microbiota class *Melainabacteria* was specifically mentioned due to its strong causality to acute necrotizing pancreatitis. This association between gut microbiota and pancreatitis may indicate biomarkers for the non-invasive identification of AP [72]. Moreover, researchers found that an alteration in gut microbiota may indicate the possibility of complications. One study showed that during the early stage of AP, higher abundances of the *Proteobacteria* phylum, *Enterobacteriaceae* family, *Escherichia*-*Shigella* genus and *Klebsiella pneumoniae* but lower abundances of the *Bifidobacterium* genus were strong indications for the occurrence of acute respiratory distress syndrome in AP patients [33]. Not only do gut microbiota show potential as biomarkers in AP; microbiome-associated substances also show potential in AP. It was reported in pancreatitis mouse models that an increased abundance of *Streptomyces*, *Turicibacter* and *Methylobacterium* was associated with pancreatic fibrosis [71]. Further experiments revealed that dysbiosis of the gut microbiota may increase CD4^+^ T cells and macrophage infiltration in pancreas tissues and modulate fibrotic progression. Since gut microbiota play a key role in AP, we can determine the severity of pancreatitis based on changes in the intestinal flora, which will determine the appropriate time to start eating and restore intestinal function as early as possible without aggravating the intestinal infection. Considering these studies, interventions based on probiotic intake could be designed to ameliorate the conditions of AP patients. Based on the relationship between the gut and AP, we can not only intervene in pancreatitis by directly interfering with the gut flora but also alleviate the inflammatory response associated with pancreatitis by influencing the pathways associated with the gut flora. AP-associated dysbiosis is often associated with activation of intracellular inflammatory pathways. The most common pathways are the TLR4-related pathway and CD14 pathway, as well as the NOD1/RIP2/NF-κB-related pathway. Inhibiting these inflammatory pathways activated by aberrant floras by designing relevant drug molecules is not a new idea for the treatment of AP.

With the further study of AP and intestinal floras, researchers have realized that intestinal microbiota are important modulatory elements in the human body. The interaction of intestinal microbiota with other organs provides multiple promising therapeutic targets. New concepts have been introduced to represent these newly discovered interactions, such as lung–gut axis, brain–gut axis and liver–gut–heart axis. AP can cause various complications, the most serious of which is acute respiratory distress syndrome (ARDS), which is diffuse damage to the lung [73]. By regulating the intestinal microbiota, reducing bacterial translocation and pathogen-associated molecular pattern (PAMP) production, the intestinal permeability and inflammatory response can be reduced [74]. This is referred to as the lung–gut axis. The protection of lung tissue can be achieved through improving intestinal barrier function, preventing pancreatic enzymes and toxins from entering the lung, balancing the immune system, inhibiting excessive pro-inflammatory factors and oxidative stress and enhancing anti-inflammatory factors and antioxidant capacity [75]. Another gut–organ axis is the gut–brain axis, which can regulate the balance of gut microbiota that may affect the function of the pancreas and reduce inflammation reactions. Through the release of neurotransmitters and hormones, the axis can inhibit the stress response, which in a way reduces the damage of pancreatitis. The protection of the integrity of the intestinal barrier by neuroimmune pathways can prevent intestinal endotoxins from entering the bloodstream and cause systemic inflammatory reactions [76]. Last but not least, in the context of the liver–gut–heart axis, the interconnectedness of different organs plays a crucial role in health. An unhealthy liver can influence the gut microbiota, which in turn can impact the cardiovascular system. Within the liver–gut axis, reducing the production of intestinal endotoxins can lower the risk of a systemic inflammatory response and multiple-organ failure caused by acute pancreatitis (AP) [77]. In the gut–heart axis, regulation of the cardiovascular system can enhance the circulatory condition of AP patients, thereby reducing the risk of stroke and heart failure associated with AP. Lastly, within the liver–heart axis, regulation of metabolic pathways can improve the nutritional status of AP patients, which in turn can reduce the risk of complications such as consumptive disorders and infections associated with AP [78].

## 6. Conclusions

In this review, we have discussed the essential role of gut microbiota in AP. We also illustrated how gut microbiota dysbiosis, bacterial translocation and molecular actors affect the pathogenesis and progression of AP and its complications. The potential therapeutic strategies that target the gut microbiota are highlighted, such as probiotics. Furthermore, we have explored the emerging concepts of gut–organ axes, providing new insights into the mechanisms and interventions of AP. However, more studies are needed to elucidate the causal relationship between gut microbiota and AP to identify the specific microbial strains and metabolites that are involved and to evaluate the safety and efficacy of microbiota-based therapies in clinical settings.

## Figures and Tables

**Figure 1 ijms-25-01159-f001:**
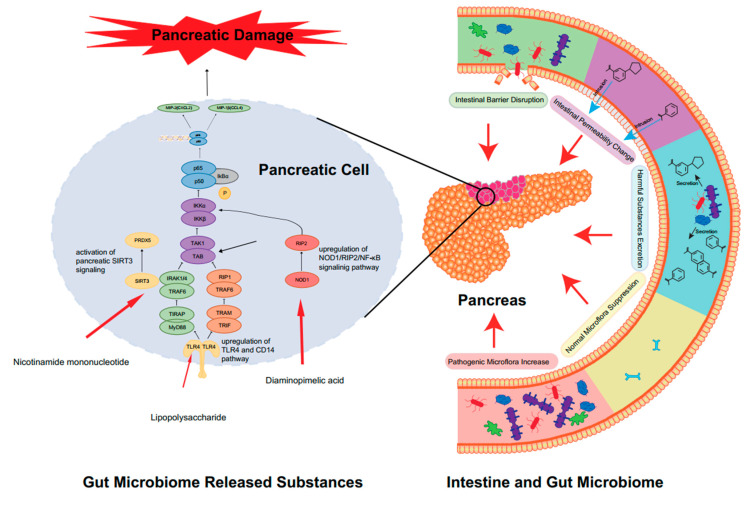
The close relationship between gut microbiota and the activation of AP. In AP, significant disorganization of the intestinal flora occurs. The first manifestation is the abnormal ratio of intestinal flora species, which is mainly characterized by an increase in harmful bacteria and a decrease in probiotics, resulting in a serious imbalance in the ratio of harmful bacteria to probiotics. Secondly, the alteration in gut flora leads to the explosion of harmful microbiome-released substances. The metabolites of harmful bacteria increase, while the beneficial metabolites of probiotics (e.g., SCFAs) decrease. Altered intestinal permeability also accelerates the progression of AP. The intestinal mucosal barrier, which can be directly and indirectly disrupted, is also affected, leading to translocation and direct invasion of the flora. These changes will cause harmful substances to accumulate in the intestine, directly stimulate the TLR4 receptor of the epithelial cells in the intestine, activate the relevant inflammatory response pathways and ultimately accelerate the destruction of pancreatic cells and aggravate AP. Some harmful substances may even enter the blood circulation directly and activate the body’s immune response, thus causing the development of AP.

**Figure 2 ijms-25-01159-f002:**
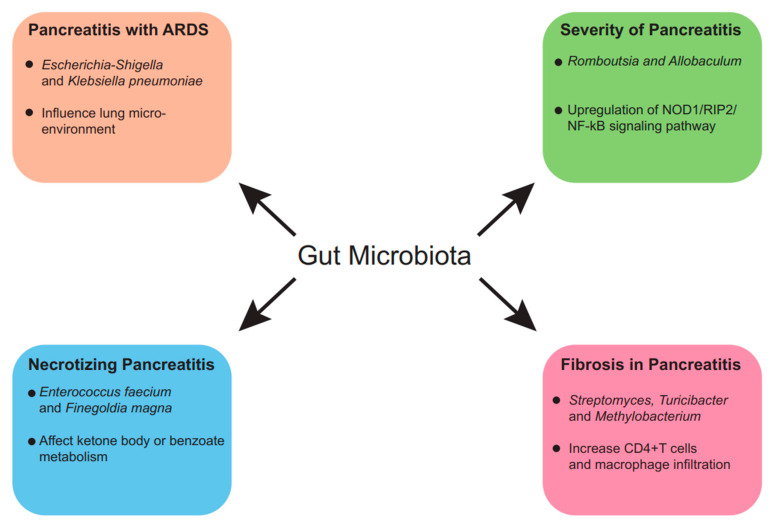
Predictive value of gut microbiota in AP. The gut microbiome is not only predictive of AP but also the severity, prognosis and comorbidities of AP, and the associations between alterations in gut microbiome and disease prognosis have underlying molecular biological mechanisms.

**Table 1 ijms-25-01159-t001:** Newly discovered molecules associated with the gut microbiome and their mechanisms in AP. (The pictures of structural formulae are from Human Metabolome Database, https://hmdb.ca/, accessed on 1 November 2023).

Molecules Correlated with AP	Potential Roles of Molecules	Year Discovered	Structural Formula
Nicotinamide mononucleotide	Activates the SIRT3-PRDX5 pathway and alleviates AP by activating pancreatic SIRT signaling [31]	2023	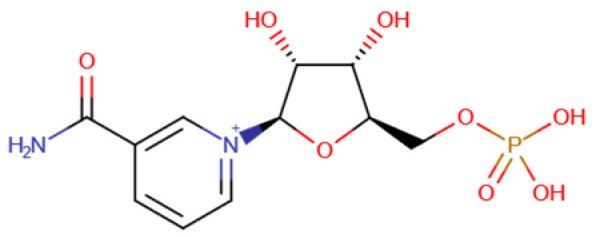
Diaminopimelic acid	Leads to the upregulation of NOD1/RIP2/NF-κB signaling pathway [30], which regulates the transcription of immune- and inflammatory-related genes and activates NF-κB in AP	2022	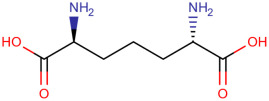
Lipopolysaccharide	Activates the Toll-like receptor 4 and CD14 pathways of innate immunity [49] and activates the NF-κB signaling pathway to trigger the inflammatory response in AP [50,51]	2022	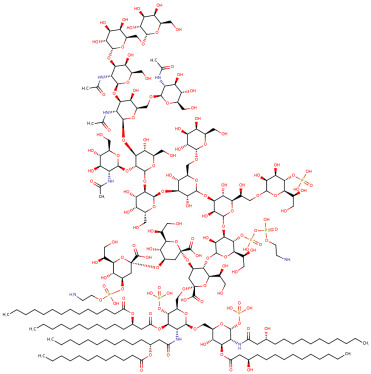
Acetate-Acetic acid	Reduces neutrophil infiltration to alleviate pancreatic inflammation [52]; inhibits the proliferation of pathogens, the anti-inflammatory effects; enhances the intestinal epithelial barrier; reduces bacterial translocation and immunomodulatory effects [6]	2022	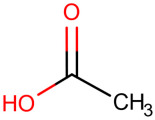
Butyric acid	Corrects intestinal microbiota imbalance, enhances intestinal mucosal barrier function and reduces bacterial translocation and infection in the intestine [53]	2021	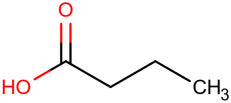
Valproic acid	Limits pancreatic recovery after pancreatitis by inhibiting histone deacetylases and prevents acinar redifferentiation programs [54]	2015	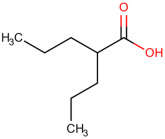

## Data Availability

Not applicable.

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
