# Peer review of "The Role of the Gut Microbiome in the Development of Acute Pancreatitis"

_ijms, 2024, doi:10.3390/ijms25021159_

Round 1

Reviewer 1 Report

Comments and Suggestions for Authors

I have added comments within body of edited ms. as shown in the attached file. Authors need to see comments inserted at strikeouts; other comments are also present. 

Authors need to be consistent in their abbreviation use. Acute pancreatitis is haphazardly abbreviated. 

Specific comment follow

Comments on the Quality of English Language

Language that is overly casual has been struck out. Terms or phrasing that are unclear have been questioned in comments inserted in text. In some places I have asked if I understood the intent. 

While the latter half of the paper is more focused the first part of the paper does not set the reader up well for what gets said. There is some confusion in what your main thesis is - by the end of the paper it seems the thesis is primarily that you might be able to use microbiota composition to diagnose, guide treatment and develop a prognosis through a "precision medicine" approach. 

The flow and integration of the pathogenesis of AP and its link to the gut was OK but rather disjointed. This is what prevented anticipation and ready acceptance of of the end section that is offering a useful application of new information related to microbiota and AP.

Reviewer 2 Report

Comments and Suggestions for Authors

Round 2

Reviewer 2 Report

Comments and Suggestions for Authors

Dear Authors, 

thank you for your responce to my comments and intoduced corrections.

Kind regrds

Author Response

Dear reviewer,

Thank you for your response to my revision.

Kind regards